# Assessment of Human SARS CoV-2-Specific T-Cell Responses Elicited In Vitro by New Computationally Designed mRNA Immunogens (COVARNA)

**DOI:** 10.3390/vaccines12010015

**Published:** 2023-12-22

**Authors:** Ignasi Esteban, Carmen Pastor-Quiñones, Lorena Usero, Elena Aurrecoechea, Lorenzo Franceschini, Arthur Esprit, Josep Lluís Gelpí, Francisco Martínez-Jiménez, Núria López-Bigas, Karine Breckpot, Kris Thielemans, Lorna Leal, Carmen Elena Gómez, Marta Sisteré-Oró, Andreas Meyerhans, Mariano Esteban, María José Alonso, Felipe García, Montserrat Plana

**Affiliations:** 1Institut d’Investigacions Biomèdiques August Pi i Sunyer (IDIBAPS), Hospital Clínic, University of Barcelona, 08036 Barcelona, Spain; iesteban@recerca.clinic.cat (I.E.); carmenpq.94.cp@gmail.cat (C.P.-Q.); lorena.usero@vhir.org (L.U.); aurrecoechea@recerca.clinic.cat (E.A.); or laleal@clinic.cat (L.L.); fgarcia@clinic.cat (F.G.); 2Laboratory for Molecular and Cellular Therapy, Department of Biomedical Sciences, Vrije Universiteit Brussel, 1090 Brussels, Belgium; lorenzo.franceschini@vub.be (L.F.); arthur.esprit@vub.be (A.E.); karine.breckpot@vub.be (K.B.); kris.thielemans@vub.be (K.T.); 3Department of Biochemistry and Molecular Biomedicine, University of Barcelona, 08028 Barcelona, Spain; gelpi@ub.edu; 4Barcelona Supercomputing Center (BSC), 08034 Barcelona, Spain; 5Institute for Research in Biomedicine (IRB Barcelona), Barcelona Institute of Science and Technology, 08028 Barcelona, Spain; fmartinez@vhio.net (F.M.-J.); nuria.lopez@irbbarcelona.org (N.L.-B.); 6Institució Catalana de Recerca i Estudis Avançats (ICREA), 08010 Barcelona, Spain; andreas.meyerhans@upf.com; 7Centro de Investigación Biomédica en Red en Cáncer (CIBERONC), Instituto de Salud Carlos III, 28029 Madrid, Spain; 8Department of Infectious Diseases, Hospital Clínic, University of Barcelona, 08036 Barcelona, Spain; 9Department of Molecular and Cellular Biology, Centro Nacional de Biotecnología (CNB), Consejo Superior de Investigaciones Científicas (CSIC), 28049 Madrid, Spain; cegomez@cnb.csic.es (C.E.G.); mesteban@cnb.csic.es (M.E.); 10Centro de Investigación Biomédica en Red de Enfermedades Infecciosas (CIBERINFEC), Instituto de Salud Carlos III (ISCIII), 28029 Madrid, Spain; 11Infection Biology Laboratory, Department of Medicine and Life Sciences, Pompeu Fabra University, 08003 Barcelona, Spain; marta.sistere@upf.com; 12Center for Research in Molecular Medicine and Chronic Diseases (CiMUS), Campus Vida, Universidade de Santiago de Compostela, 15782 Santiago de Compostela, Spain; mariaj.alonso@usc.es

**Keywords:** SARS-CoV-2, mRNA, vaccine, T-cell, dendritic cells

## Abstract

The COVID-19 pandemic has brought significant changes and advances in the field of vaccination, including the implementation and widespread use of encapsidated mRNA vaccines in general healthcare practice. Here, we present two new mRNAs expressing antigenic parts of the SARS-CoV-2 spike protein and provide data supporting their functionality. The first mRNA, called RBD-mRNA, encodes a trimeric form of the virus spike protein receptor binding domain (RBD). The other mRNA, termed T-mRNA, codes for the relevant HLA I and II spike epitopes. The two mRNAs (COVARNA mRNAs) were designed to be used for delivery to cells in combination, with the RBD-mRNA being the primary source of antigen and the T-mRNA working as an enhancer of immunogenicity by supporting CD4 and CD8 T-cell activation. This innovative approach substantially differs from other available mRNA vaccines, which are largely directed to antibody production by the entire spike protein. In this study, we first show that both mRNAs are functionally transfected into human antigen-presenting cells (APCs). We obtained peripheral blood mononuclear cell (PBMC) samples from three groups of voluntary donors differing in their immunity against SARS-CoV-2: non-infected (naïve), infected-recovered (convalescent), and vaccinated. Using an established method of co-culturing autologous human dendritic cells (hDCs) with T-cells, we detected proliferation and cytokine secretion, thus demonstrating the ability of the COVARNA mRNAs to activate T-cells in an antigen-specific way. Interestingly, important differences in the intensity of the response between the infected-recovered (convalescent) and vaccinated donors were observed, with the levels of T-cell proliferation and cytokine secretion (IFNγ, IL-2R, and IL-13) being higher in the vaccinated group. In summary, our data support the further study of these mRNAs as a combined approach for future use as a vaccine.

## 1. Introduction

As of March 2023, there have been at least 760 million confirmed COVID-19 cases and 7 million COVID-related deaths worldwide [1], with a devastating impact on the world’s human population and economy. Since the first cases were reported, the knowledge and understanding of SARS-CoV-2 biology and the immune response elicited by the virus has progressed at an unprecedented pace [2]. Within a year of SARS-CoV-2 being identified, multiple vaccines were available on the market. Among these, two mRNA-based vaccines, BNT162b2 (Comirnaty^®^; BioNTech and Pfizer) and mRNA-1273 (Spikevax; Moderna), were the most successful at preventing symptomatic and severe disease [3]. Both vaccines consist of a strain of mRNA encoding the SARS-CoV-2 spike protein stabilized in its pre-fusion conformation and wrapped in a lipid nanoparticle (LNP). The LNP protects the mRNA from degradation by nucleases while facilitating cell internalization by antigen-presenting cells (APCs), which produce and present the antigen [4]. In addition, LNPs have strong adjuvant activity due to their ionizable lipid component, which increases the efficacy of these vaccines [5]. 

After two shots of the BNT162b2 or mRNA-1273 vaccine, high titers of neutralizing antibodies are produced [6,7]. In addition, these vaccines generate robust CD4 and CD8 T-cell responses, which help both to mitigate disease severity and to clear the infection [8,9]. Importantly, some studies performed with animal models have found that T-cell immunity could be enough to confer protection against SARS-CoV-2 when there are suboptimal [10] or even undetectable [11] levels of neutralizing antibodies. Indeed, a relevant feature of mRNA vaccines is their more potent induction of the cytotoxic CD8 T-cell response, as compared with traditional vaccines such as inactivated viruses [12] or those which are protein-based [13]. 

Here, we describe the activity of a new mRNA-based immunogen (COVARNA) against the original (Wuhan) strain of the SARS-CoV-2 virus. This immunogen consists of a mixture of two different mRNAs. The first mRNA, named RBD-mRNA, codes for a trimeric soluble form of the spike receptor binding domain (RBD). The RBD was selected as the antigen instead of the whole spike protein because this protein domain is the main target of neutralizing antibodies (NABs) [14]. The other mRNA is a T-cell activating immunogen (T-mRNA) that codes for some of the T-cell peptides most recognizable by the HLA system in the Spanish population, according to selection by computational methods. The criteria for selecting the T-cell epitopes included in the T-mRNA were broad HLA coverage, divergence from human proteome to avoid autoimmunity, and sequence conservation to limit viral mutational escape. 

In this study, we analyzed the ability of these new mRNA immunogens to activate SARS-CoV-2-specific T-cells in vitro. To this end, we first confirmed the correct expression of both immunogens on human dendritic cells (hDCs). We then analyzed the activation of T-cells in three groups of individuals: non-infected (naïve), SARS-CoV-2-infected, and recovered (convalescent), and vaccinated.

## 2. Materials and Methods

### 2.1. mRNA Design and Production

We designed two different mRNAs, called RBD-mRNA and T-mRNA (Figure 1). RBD-mRNA encodes a trimeric RBD of the SARS-CoV-2 spike protein (aa: 330-532), modified by adding a T4 fibritin-derived fold on the trimerization domain to increase its immunogenicity. 

The T-mRNA multiepitopic immunogen codes for a polypeptide chain containing 14 HLA class I epitopes, each separated by three alanine residues to optimize cleavage by the proteasome, and 5 HLA class II epitopes connected by a highly flexible linker GPGPG from the spike protein. The construct was designed computationally according to the following criteria: We predicted the binding affinity of all potential 9-mers of the spike protein of the original SARS-CoV-2 Wuhan strain against the most frequent HLA-I alleles in Caucasian population. The MHCflurry2.0 tool was used to predict HLA-I presentation. Peptide–HLA-I pairs with predicted binding affinity lower than 500 nM were considered as binders. Similarly, all potential 12–21-mers of the spike protein of the original SARS-CoV-2 strain were virtually screened against the most frequent HLA-II alleles using MHC nuggets. Peptide–HLA-II pairs with predicted binding affinity lower than 500 nM were considered as binders.Peptides with at least one predicted HLA-I allele or HLA-II allele were pre-selected.Pre-selected peptides in (2) that had a perfectly matching sequence in the human canonical proteome were discarded to avoid autoimmunity.Pre-selected peptides bearing more than 6 mutations (i.e., lowly conserved across spike viral sequences) were discarded to avoid immune escape. The only exception was LPLVSSQCV (7 observed mutations), as this epitope is considered one of the most promiscuous within the entire spike protein. GISAID (https://gisaid.org/; URL accessed on 2 June 2020) was used as a source to annotate the number of mutations in the SARS-CoV-2 spike protein.We then developed a 14-amino acid polypeptide T-cell construct that maximized HLA-I coverage in the Caucasian population by sampling all possible combinations of 14 pre-selected peptides. The estimated HLA-I Caucasian coverage for these combinations was calculated using the IEDB Population Coverage 1.0.1 tool. A similar approach was taken to design a 5-amino acid polypeptide T-cell HLA-II construct using pre-selected HLA-II peptides from the previous steps.

T-mRNA also codes for the trans-membrane and cytosolic domains of the DC-LAMP protein [15]. The inclusion of this sequence on therapeutic mRNAs has been demonstrated to improve the presentation of antigens by HLA class II molecules, as it directs the synthesis of the polypeptide chain to the endoplasmic reticulum (ER) while also improving HLA class I presentation [16]. This HLA class II targeting sequence has been used with a safety profile in different clinical trials [17,18]. A TAG version of T-mRNA including two detectable epitopes, M3/DP4 and p53/A2, was also produced. In the present study, when both mRNAs (RBD-mRNA + T-mRNA) are used in combination, we refer to them as COVARNA mRNAs.

### 2.2. Transfection of mRNA to K562-A2^+^ Cells and CD8 T-Cells by Electroporation

Transfection of T-mRNA with TAG into the K562-A2^+^ cell line and p53 TCRα and TCRβ-chain mRNA into CD8 T-cells was performed by electroporation. Cells were extensively washed in serum-free OptiMEM (Gibco-Thermo Fischer Scientific, Waltham, MA, USA). The electroporation was performed in 200 μL of OptiMEM medium in a 4 mm electroporation cuvette (Cell Projects, Harrietsham, UK) using the following parameters: square wave pulse, 500 V, 2 ms, 1 pulse for K562-A2^+^ cells; and square wave pulse, 500 V, 5 ms, 1 pulse for T-cells, using the Gene Pulser Xcell device (Bio-Rad, Hercules, CA, USA). p53 TCRα and TCRβ-chain mRNA (5 μg each, 4/10^6^ cells) was electroporated into CD8^+^ T-cells [19]. T-mRNA with TAG (5 μg each, 2/10^6^ cells) was electroporated in K562.

### 2.3. mRNA Translatability Test by Antigen Presentation Assay

Monocyte-depleted PBMCs were thawed and cultured for 1 day in the presence of 25 IU/mL of IL-2. CD8^+^ T-cells were then isolated from monocyte-depleted PBMCs by magnetically activated cell sorting (MACS) using positive selection with human anti-CD8 microbeads, according to the manufacturer’s instructions (Miltenyi Biotec, Bergish Gladbach, Germany). After isolation, CD8^+^ T-cells were rested for 2 h and then electroporated with the TCR p53 mRNA. The K562-A2^+^ cells were electroporated with the TAG T-mRNA. One hour after electroporation, the CD8^+^ T-cells and the K562-A2^+^ cells were co-cultured at a 1:1 ratio in the presence of IL-2 (25 IU/mL). Cells were plated in triplicate in 96-well round-bottom plates in IMDM supplemented with 1% human AB serum (200 μL/well) for 24 h at 37 °C and 5% CO_2_. Supernatants from the co-cultured cells were collected to quantify IFNγ in ELISA (Thermo Fisher Scientific, Waltham, MA, USA) according to the manufacturer’s instructions. An STD p53 mRNA was used as a positive control [19].

### 2.4. Samples and Study Participants

Samples for testing the detection of immunogens on hDCs were obtained from blood donors (buffy coats) through the Barcelona Blood and Tissue Bank (BST). 

Samples for the co-culture experiments were extracted from health personnel working at the Hospital Clinic de Barcelona—IDIBAPS—all of whom signed written informed consent. Health personnel were divided into three groups based on their status of immunity against SARS-CoV-2: (1) naïve (non-infected) individuals (*n* = 4) who routinely tested negative for SARS-CoV-2 by PCR and for antibodies; (2) convalescent individuals (*n* = 6) who tested positive for SARS-CoV-2 by PCR during the first half of 2020, and from whom samples were extracted between two and six months after testing positive; and (3) vaccinated individuals (*n* = 5) who received two doses of BNT162b2 or mRNA-1273 between January 2021 and September 2021, and from whom samples were extracted between two and six months after vaccination. The median age of donors (*n* = 15) was 37.9 years (SD ± 10.4) and the sex ratio was 60% male, 40% female. The study was evaluated and approved by the Committee of Ethics and Clinical Research of the Hospital Clinic i Provincial de Barcelona, Spain (HCB/2020/0387).

### 2.5. Generation of Monocyte-Derived Dendritic Cells

Peripheral blood mononuclear cells (PBMCs) were isolated from a buffy coat or from a 100 mL sample of EDTA-treated venous blood by a Ficoll-Hypaque density gradient. PBMCs were incubated for 2 h at 37 °C and 5% CO_2_. After incubation, monocytes adhered to the flask and the non-adherent fraction of cells was removed by three washes with warm PBS. Monocytes were cultured for 6 days with 1000 IU/mL of recombinant IL-4 and 1000 IU/mL granulocyte macrophage colony-stimulating factor (GM-CSF) (Prospec Bio, Rehovot, Israel). After 6 days of culture, monocytes were transformed into immature human dendritic cells (ihDCs).

### 2.6. Transfection by Electroporation of Human DCs with mRNA

Immature human dendritic cells (ihDCs) were collected and washed twice with PBS (2000 rpm, 5 min). Cells were then re-suspended in OptiMem medium (Gibco/Thermo Fischer Scientific, Waltham, MA, USA) at a final concentration of 20 × 10^6^ cell/mL. For each condition, 200 μL of the cell suspension was placed into a 0.4 cm gap sterile electroporation cuvette. Next, 20 μg of mRNA [10 μg RBD-mRNA + 10 μg T-mRNA or 20 μg of GFP-mRNA (StemMACS (TM) eGFP mRNA, Milteny Biotec, Bergisch Gladbach, Germany)] was mixed with 200 μL of cell suspension. Electroporation was performed using the Gene Pulser II electroporation system (Bio-Rad, Hercules, CA, USA) and the following parameters: 300 V, 150 µF, and 800 Ω. ihDCs without mRNA were electroporated as negative control (mock cells). After transfection, cells were placed in tubes with 2 mL of RPMI medium with 10% fetal bovine serum (FBS) and gentamicin (50 μg/mL) and rested for 2 h (37 °C in 5% CO_2_). Cells were then counted and placed on a p96 U-well bottom plate at a concentration of 6 × 10^5^ cell/mL in 200 μL of volume. Maturation cocktail (IL-1β, TNFα, IL-6, and PGE2) was added to all wells.

### 2.7. Co-Culture of hDCs with T-cells: Proliferation and Cytokine Secretion

hDCs were obtained as described above from a 100 mL blood extraction and electroporated either with COVARNA mRNAs (10 μg RBD-mRNA + 10 μg T-mRNA) or without mRNA (mock cells), as previously described. After electroporation, cells were rested for 3 h in complete autologous medium to recover from electroporation. During those 3 h of resting, CEF peptide pool (Mabtech, Nacka Strand, Sweden) was added to some mock cells to be used as a positive control for the cytokine secretion assay (CEF was used at a concentration of 2 μg/mL per individual peptide). After the 3 h of resting, maturation cocktail was added to electroporated hDCs and cells were cultured for 24 h (37 °C and 5% CO_2_) to obtain mature hDCs. The next day, a second blood extraction of 40 mL from the same donor was performed and the isolated PBMCs were depleted from monocytes by plate adhesion. Monocyte-depleted PBMCs were then stained with CFSE (Invitrogen-Thermo Fischer Scientific, Waltham, MA, USA) and co-cultured with electroporated hDCs. PHA (Sigma-Aldrich, Burlington, MA, USA) at a concentration of 35 μg/mL was added to the co-culture as a positive control for T-cell proliferation. On the last day of co-culture (12 h before obtaining the supernatants), a mix of overlapping peptides covering the S1 subunit of the SARS-CoV-2 spike protein (PepMix SARS-CoV-2 spike glycoprotein PM-WCPV-S-2, JPT Peptides, Berlin, Germany) was added (20 μg/mL of whole mix) to some wells to stimulate cytokine secretion from virus-specific T-cells. On day 6 of co-culture, the supernatant was collected to analyze the presence of cytokines and chemokines in the medium, and cells were recovered and stained to assess T-cell proliferation by flow cytometry analysis.

### 2.8. Flow Cytometry

Intracellular expression of the RBD protein after electroporating the RBD-mRNA on hDCs was detected using a rabbit polyclonal antibody against the SARS-CoV-2 RBD (Cat: 40592-T62, Sino-Biological, Beijing, China) as a primary antibody, and a goat anti-rabbit IgG conjugated with A647 (A32733, InVitrogen-Thermo Fischer Scientific, Waltham, MA, USA) as a secondary antibody. Cell viability was analyzed using a LIVE/DEAD fixable Near-IR Dead Cell Stain Kit (InVitrogen-Thermo Fischer Scientific, Waltham, MA, USA). In the T-cell proliferation assay, cells were stained with the following mAbs: CD3-PerCp (clone: SK7), CD4-BV421 (clone: RPA-T4), and CD8-BV510 (clone: SK1), all from BD Pharmingen. In all cases, samples were acquired with the FACS Canto II flow cytometer (BD Biosciences, New Jersey, NY, USA) and analyzed with FlowJo 10 software (Tree Star, Ashland, OR, USA).

### 2.9. Analysis of Cytokines and Chemokines

Cytokines and chemokines in the supernatant of co-cultures were analyzed with the Luminex Cytokine Human Magnetic 25-Plex Panel (InVitrogen-Thermo Fischer Scientific, Waltham, MA, USA) according to the manufacturer’s protocol. This kit allows for the simultaneous detection of the following cytokines and chemokines: GM-CSF, IFNα, IFNγ, IL-1β, IL-1RA, IL-2, IL-2R, IL-4, IL-5, IL-6, IL-7, IL-8, IL-10, IL-12 (p40/p70), IL-13, IL-15, IL-17, TNFα, Eotaxin, IP-10, MCP-1, MIG, MIP-1α, MIP-1β, and RANTES.

### 2.10. Statistical Analysis

Statistics were performed with GraphPad Prism 8.0 (GraphPad Software, San Diego, CA, USA), using the Mann–Whitney test to compare differences between two different groups or the Wilcoxon matched-pairs test when comparing paired samples. In the results, statistical significance is indicated as follows: ns = non-significant; * *p* < 0.05; ** *p* < 0.005; *** *p* < 0.001.

## 3. Results

### 3.1. B-Cell Immunogen Detection

To test the capacity of the RBD-mRNA to express the RBD protein, the hDCs were transfected with RBD-mRNA to detect the protein expression on that pivotal set of antigen-presenting cells. The results showed that RBD protein expression was detectable on the hDCs, with more than 50% of the cells expressing RBD at 6 h after electroporation (60.7% ± 13.1%; *p* = 0.0023) (Figure 1A,B). At 24 h, the percentage of cells expressing RBD decreased by half (27.3% ± 14.6; *p* = 0.036), and after 48 h, the RBD expression was at ten percent (10% ± 13.2%; ns) (Figure 1A,B). On the hDCs electroporated with a control eGFP mRNA, RBD expression was not detected, whereas eGFP expression was clearly detected at all time points, with the highest percentage of eGFP-positive cells being observed at 24 h post-transfection (34.4% ± 22%) (Figure 1B). 

As the commercial mRNA vaccines contain the modified nucleotide N1-methylpseudouridine (m1Ψ) [20,21], the same experiments were also performed comparing an m1Ψ modified version of RBD-mRNA with the non-modified version (Appendix A). Similar transfection efficiencies were observed with the m1Ψ modified and non-modified version of RBD-mRNA. This lack of differences between the modified and non-modified RBD-mRNA could be due to the method of transfection, as with electroporation, the mRNA reaches the cytosol without transiting the endocytic pathway, avoiding recognition by Toll-like receptors and degradation [22]. Therefore, for practical reasons, the following experiments were carried out solely with the non-modified mRNA.

### 3.2. T-Cell Immunogen Detection 

As T-mRNA does not code for a tridimensional structure that could be easily targeted with an antibody to confirm its translation, a TAG sequence was included in the T-mRNA. The TAG codes for an epitope (LLGRNSFEV) of the p53 protein that is presented by HLA-A2 molecules and can be recognized by a specific TCR (p53 TCRα- and TCRβ-chain) [19]. T-mRNA with the p53/A2 TAG was transfected into K562 HLA-A2^+^ cells and then co-cultured for 24 h with CD8 T-cells that had been previously transfected with a specific TCR against the TAG. The production of IFNγ in the supernatant was then evaluated by ELISA.

As expected, when the K562-A2^+^ cells were transfected with a control eGFP mRNA or with T-mRNA without the p53 TAG, there was no detection of IFNγ, whereas transfection with the TAG version of the T-mRNA and the positive control (SDT p53 mRNA) induced a robust release of IFNγ by CD8 T-cells, confirming the correct translation of the T-mRNA (Figure 2).

### 3.3. T-Cell Proliferation Induced In Vitro by COVARNA Transfected Autologous hDC

After confirming that both mRNAs were correctly translated, we studied their ability to stimulate T-cells in presentation assays. To this end, three different groups of donors were selected based on their status of immunity against SARS-CoV-2; non-infected (naïve), infected (convalescent), and vaccinated. Using an established method of co-culturing autologous hDCs with monocyte-depleted PBMCs [23,24], we were able to detect antigen-specific proliferation and cytokine secretion. The hDCs were electroporated with the COVARNA mRNAs (COVARNA-transfected DCs) and later co-cultured with CFSE-stained autologous monocyte-depleted PBMCs. The culture was maintained for 6 days, at which point the proliferation of T-cells was analyzed by flow cytometry (Figure 3A,B). 

The results showed that the COVARNA-transfected hDCs induced a statistically significant increase in the proliferation of both the CD4 and CD8 T-cells in the mRNA-vaccinated individuals (CD4 T-cell: 17.9% ± 12.7%; CD8 T-cell: 14.6% ± 10.4%) as compared with the non-infected (naïve) individuals (CD4 T-cells: 0.7% ± 0.5%; CD8 T-cells: 1.1% ± 1.3%) (*p* = 0.0314 CD4 T-cells; *p* = 0.0314 CD8 T-cells). Among the convalescent individuals, there was also a slight increase in the proliferation of both the CD4 and CD8 T-cells (CD4 T-cell: 2.5% ± 2.5%; CD8 T-cell: 5.5% ± 6.7%) in comparison with the non-infected individuals (CD4 T-cells: 0.7% ± 0.5%; CD8 T-cells: 1.1% ± 1.3%), although, in this case, the difference was not statistically significant (Figure 3B). As expected, the cells stimulated with PHA showed similar levels of T-cell proliferation across the study groups. 

### 3.4. Cytokine Secretion in Response to COVARNA mRNA

The secretion of different cytokines and chemokines in the medium was evaluated 6 days after establishing the co-culture. The results showed that in the vaccinated donors, the COVARNA-transfected hDCs induced a statistically significant increase in IL-2R (*p* = 0.0317), IL-13 (*p* = 0.0317), and TNFα (*p* = 0.0159) compared with the non-infected donors (Figure 4A–D). IFNγ (134 pg/mL ± 130) and IL-6 (32 pg/mL ± 26.6) (Figure 4E,F), were also increased in the vaccinated compared to non-infected donors (IFNγ 26 pg/mL ± 40; IL-6 1.3 pg/mL ± 2.5), but the differences are not significant. In the infected individuals (convalescents), no significant differences were found in any of the cytokines, compared to the non-infected individuals (Figure 4F), but two individuals showed a clear IFNγ response (Figure 4E). IL-2 secretion was undetectable in any of the three groups (Figure 4F). The analysis of the other cytokines and chemokines (Appendix A), revealed significant increases in GM-CSF (*p* = 0.0159), RANTES (CCL5) (*p* = 0.0159), MIP-1α (CCL3) (*p* = 0.0317), and MIP-1β (CCL4) (*p* = 0.0159) in the vaccinated donors compared with the non-infected (naïve) individuals.

To better characterize the specific T-cell activation induced by the COVARNA-transfected hDCs, a pool of overlapping peptides covering the spike S1 subunit was added to the culture 12 h before collecting the supernatant. In all three groups of donors (non-infected [naïve], infected-recovered [convalescent], and vaccinated) the addition of the overlapping peptides increased the release of IFNγ only when the COVARNA-transfected hDCs were used, and not with the mock hDCs (Figure 5A), with the highest increase in IFNγ being detected in the vaccinated group (non-infected: 42.4 pg/mL ± 84.9 to 142.4 pg/mL ± 275; infected: 8 pg/mL ± 14 to 203.4 pg/mL ± 253; vaccinated: 8.6 pg/mL ± 10.9 to 271.2 pg/mL ± 198.2). We also detected an increase in the IL-2 levels when using the COVARNA-transfected hDCs but only in the vaccinated group (16.3 pg/mL ± 12.8 to 39.2 pg/mL ± 47.3), and not in either the infected (convalescent) or non-infected (naïve) groups (Figure 5B).

## 4. Discussion

The COVID-19 pandemic has given a definitive boost to mRNA vaccine technology, with mRNA vaccines superseding traditional vaccines in multiple aspects [25,26]. Here, we present two new mRNAs (RBD-mRNA and T-mRNA) containing antigenic parts of the SARS-CoV-2 spike protein, and we report preclinical data supporting the ability of both mRNAs (COVARNA mRNAs) to induce an in vitro specific T-cell response. 

The RBD-mRNA codes for a trimeric form of the RBD from the original (Wuhan) strain of SARS-CoV-2. As for the T-mRNA, this codes for some relevant HLA-I and HLA-II T-cell epitopes of the spike protein. The rationale for using two antigenic sequences is that RBD-mRNA will serve as the main source of antigen for the generation of neutralizing antibodies and T-cells, while T-mRNA will provide additional epitopes to support the activation of specific T-cells. Both mRNAs, RBD-mRNA and T-mRNA, were designed to be separated into two different mRNA molecules, as having just one larger chain could potentially compromise the mRNA stability [27]. This design differs substantially from other available mRNA-based vaccines, which are largely based on the entire spike [8,21]. 

As a first step in assessing the functionality of the RBD-mRNA, we decided to test its ability for expressing the RBD protein on human antigen-presenting cells (APCs). Following the electroporation of the hDCs with RBD-mRNA, the protein was rapidly (6 h) detectable, after which it progressively vanished (at 24 and 48 h). 

To detect the presentation of the T-mRNA TAG by K562-A2^+^, the isolated CD8 T-cells were transfected with a TCR mRNA specific for the epitope in the TAG and then co-cultured with the K562-A2^+^ cells. The detection of IFNγ in the supernatant of the co-cultures of the K562-A2^+^ cells with CD8 T-cells confirmed that the polypeptide chain encoded by the T-mRNA was expressed, processed into small peptides by the proteasome, and presented by HLA-A2 molecules, thus verifying the functionality of the T-mRNA.

In the co-cultures, as expected, the results showed that in the non-infected (naïve) group, proliferation was minimal compared with the infected-recovered (convalescent) and vaccinated individuals [28]. The secretion of IFNγ and IL-2R, two cytokines associated with antigen-specific activation [29], was also minimal in the non-infected group. By contrast, in the infected-recovered (convalescent) and vaccinated donors, there was an increased proliferation of CD4 and CD8 T-cells and an increased secretion of IFNγ. The less intense proliferation and cytokine secretion observed in the infected-recovered group in comparison with the vaccinated group could be explained, at least in part, by the fact that an infection has to generate responses against all the proteins of a virus, and hence, the immune system has to divide resources among multiple targets [30]. For example, after infection in naïve individuals, the most immunodominant CD8^+^ T-cell responses (B7/N_105_^+^CD8^+^ T-cells) are directed against peptides from non-spike regions (N protein, M protein, and ORF1ab) [31,32,33]. By contrast, with BNT162b and mRNA-1273 vaccines, the immune system only targets a single protein (S), and therefore, more resources are available to mount a response against the spike. As all antigens contained in the COVARNA mRNAs are from the spike, one would expect to observe more robust responses in the vaccinated group. As regards IL-2 secretion, this was undetectable in any of the three groups. This could be because IL-2 acts as an autocrine and paracrine mediator that is used by T-cells to proliferate [34,35], and therefore, after 6 days of culture, all the IL-2 released following presentation was likely consumed by the proliferating T-cells. IL-13 was also increased in the vaccinated group compared with the infected (convalescents) and naïve donors, but other Th2 cytokines such as IL-4 or IL-5 were not up-regulated. This is in line with similar results obtained by Vogel et al. [20], in which splenocytes from mice that had been vaccinated with BNT162b2 secreted IL-13 after re-stimulation with SARS-CoV-2 spike overlapping peptides, but not other Th2 cytokines such as IL-4 or IL-5.

The secretion of chemokines was also detected after 6 days of culture of COVARNA-transfected hDCs with T-cells. Notably, the levels of MIP-1β (CCL4), a chemokine that is secreted by T-cells [36,37], were augmented in infected-recovered and vaccinated donors, indicating T-cell activation. By contrast, CXCL9 and CXCL10, two chemokines that are secreted by other types of cells (monocytes, endothelial cells, and fibroblasts) [38] remained unchanged across the three groups of donors. 

We also found that the addition of peptides from the spike protein 12 h before collecting the medium increased the release of IFNγ in infected-recovered (convalescent) and vaccinated donors. This increase in IFNγ only occurred when the COVARNA-transfected DCs were used, and not with mock DCs, indicating that COVARNA-transfected DCs have augmented the few SARS-CoV-2 specific T-cells present at the beginning of the culture. In non-infected-donors, no IFNγ response was detected. In the case of the individual in the non-infected group with a clear IFNγ response to the peptides, this effect can likely be attributed to a non-detected infection or to cross-reactivity with other coronaviruses [39], rather than to a priming effect of the COVARNA mRNAs 6 days earlier. Note also that the addition of peptides only induced an increase in IL-2 in the vaccinated group, and not in the infected (convalescent) individuals. This lack of IL-2 response when adding the peptides to the infected-recovered (convalescent) group could be attributed to the fact that, even in vaccinated donors, the amounts of IL-2 detected were very low (0–50 pg/mL), close to the limit of detection for the technique used. With less intense activation, as seems to be the case that for infected-recovered (convalescent) individuals, it is plausible that the tiny amounts of IL-2 produced in response to the peptides could not be detected.

Finally, although the data presented here clearly demonstrate the ability of RBD-mRNA and T-mRNA to induce an in vitro T-cell response, one of the limitations of our study is that it lacks data on B-cell responses. This issue, along with the need to provide data on the functionality of the two mRNAs with a nano-carrier, must be addressed in future research. Another important limitation of the study concerns the number of subjects, which may limit the statistical power. At present, however, it is almost impossible to obtain additional samples from the three groups of donors used in the study, due to the current epidemiological situation and the extensive coverage of vaccinated individuals. 

## 5. Conclusions

In summary, the data presented here demonstrate that COVARNA mRNAs are functional and express their antigenic content when transfected in APCs. Furthermore, COVARNA-transfected APCs can activate in vitro specific CD4 and CD8 T-cell responses against SARS-CoV-2 antigens. This activation depends mainly on the subject’s status of immunity against SARS-CoV-2, with the highest responses observed in previously vaccinated subjects, suggesting that COVARNA mRNAs could potentially function as a booster dose strategy. One of the main advantages of working with an mRNA-based platform is that once it is optimized, it can easily be adapted to other variants by simply changing the sequences [40]. This means that the immunogens presented in this study, which contain the sequence of the original (Wuhan) strain of SARS-CoV-2, represent a proof-of-concept and could easily be adapted to new variants with little effort.

## Data Availability

The datasets generated during and/or analyzed during the current study are available from the corresponding author on reasonable request due to privacy and ethical reasons.

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
