# Peer review of "Assessment of Human SARS CoV-2-Specific T-Cell Responses Elicited In Vitro by New Computationally Designed mRNA Immunogens (COVARNA)"

_vaccines, 2023, doi:10.3390/vaccines12010015_

Round 1

Reviewer 1 Report

Comments and Suggestions for Authors

1. The study designed two mRNA vaccines, of which the T-mRNA vaccine contains 19 T-cell epitopes. Although the author gave the screening criteria for epitopes in the method, the criteria were too haphazard, and the method did not mention the specific source of epitopes, whether the epitopes had been verified by experiments, and which HLA molecules the epitopes specifically covered. 

2. In Figures 3 and 4, why were the values on the negative control (Mock hDCs) subtracted from the graphs, and have you considered including this portion of the results in the supplementary materials? It should be critical.

3. In the discussion, there have been multiple mentions of certain index not being detected, which may be related to the six-day in vitro culture of DC cells: “This fact could be explained because IL-2 acts as an autocrine and paracrine mediator used by T-cells to proliferate, therefore after 6 days of culture all the IL-2 released after presentation has probably been consumed by the proliferating T-cells; On non-infected-donors, no IFNγ response is detected, except for one individual, and therefore COVARNA transfected DCs are not enough to prime T-cells in just six days of in vitro culture.” Therefore, we wonder how the six-day culture period was determined and whether any attempts have been made to optimize the culture duration.

4. The article mentioned the addition of peptides only induced an increase of IL-2 on the vaccinated group, but not on infected (convalescent). With a less intense activation, as seems to be the case for infected-recovered (convalescent) individuals, it is plausible that the tiny amounts of IL-2 produced in response to the peptides could not be detected. Have you ever considered optimizing the detection method to detect IL-2 secretion?

5. In Result 3.4, the MS told that IL-2 secretion was undetectable any of three groups in Figure.4D, but it seems that figure 4F did represent IL-2 secretion.

6. The study designed two mRNA vaccines, of which the T-mRNA vaccine contains 19 T-cell epitopes. Although the author gave the screening criteria for epitopes in the method, the criteria were too haphazard, and the method did not mention the specific source of epitopes, whether the epitopes had been verified by experiments, and which HLA molecules the epitopes specifically covered.

Comments on the Quality of English Language

There were typos and misleading expression. Moderate editing of English language required 

Reviewer 2 Report

Comments and Suggestions for Authors

Strengths: The vaccine approach in this manuscript seems novel and the results seem sufficient to merit publication. The discussion does a good job of clarifying the rationale for the experimental design and offers plausible explanations for the results when they were not what was expected.

Weaknesses: The results are still at a preliminary stage, in that the mRNAs are not encapsidated but are instead electroporated. Also evidence of immune stimulation only occurred in cells from people with previous exposure to SARS-CoV-2, not naive individuals, which is not promising for a vaccine.

Suggestions for improvement:

1) Line 78-81: Mischaracterizes the meaning of reference 6. The RNA in unmodified mRNA vaccines induces type I interferons but it is not an adjuvant - it prevents the vaccines from working by blocking translation. 

2) The organization of the Methods section is difficult to understand, and information is missing at times.

Section 2.1: All of the constructs should be described including the TCRa and TCRb constructs. What is DC LAMP and M3DP4 in scheme I?

Line 145: what is an STD p53 mRNA?

3) Results: Integrate parts of the discussion (lines 380-457) into the results section to make it easier to follow the reasoning for the experimental approach. 

Figure IB. The statistics do not seem appropriate/correct - comparing RBD-mRNA to mock transfection? No similar stats on eGFP-mRNA?

Figure 2: No statistical analysis? Is it STD p53 or SDT p53 - both are in the text.

4) Section 3.4 is very hard to follow. Perhaps a table would be better than a figure? Or just rewrite to make it easier to match with the figure. Is Fig 4D correct in line 323? Text says IL-2, graph says IL-6.

5) can shorten the discussion by moving much to results (see above). Also reduce the repetition between background, results and discussion.

Comments on the Quality of English Language

The word "on" is used throughout in ways that the word "in" is usually used. For example:

line 293

"....increase of proliferation on both CD4 and CD8 T-cells" should be "....increase of proliferation in both CD4 and CD8 T-cells"

line 343

  1.  
  2. "On the three groups of donors" should be "In the three groups of donors"

Reviewer 3 Report

Comments and Suggestions for Authors

In their work, Ignasi Esteban and colleagues study the ability of their proprietary COVARNA immunogens to induce in-vitro immune responses. The authors show that upon electroporation with their construct, human monocyte derived dendritic cells are able to induce proliferation of T cells from autologous donors previously vaccinated against SARS-CoV-2 which suggests that the DCs present the RBD encoded in the electroporated mRNA.

The authors do not show how the T-mRNA is different compared the RBD-mRNA alone, the only experiment where T-mRNA is tested refers to p53/A2 TAG – mediated effect in absence of RBD-mRNA. It would be interesting to see the comparison.  Because of this, the added value of T-mRNA is difficult to justify.

How do the authors explain lack of IFNy secretion in vaccinated donors in CD8+T cell cocultures stimulated with S1 peptide mix (Figure 5A – mock electroporated hDCs)? Shouldn’t one expect for the Covid-19 specific CD8+ T cells to be there and react to the peptide pool? Same goes for the convalescent samples.

Minor comments:

Line 51:

functional, expressing their antigenic content when transfected on human antigen presenting cells should be: transfected into human antigen presenting cells

Line 53:

groups of voluntary donors differing on their immunity status against SARS-CoV-2

should be: differing in their immunity

Line 59:

on the vaccinated group

should be: in the vaccinated group

Line 70:

available in the market

should be: available on the market

Line 80-81:

and other relevant cytokines for the antiviral response

should be: and other cytokines relevant for the antiviral response

In the materials and methods section timeline for T cell electroporation should be mentioned – how many days post isolation were the cells electroporated, were the cells activated, was IL-2 used?

In the line 129 OptiMEM is described as “(Life Technologies, Belgium)”. In the line 174 “OptiMem medium (Gibco/Thermo 174 Fischer Scientific, USA)” is used instead. Could the authors make it uniform?

Line 208 – it is not clear what the authors mean by B-cell construct at this point. It should be rephrased “electroporated construct”.

The text says that RBD was not detectable in hDCs 48 hours after electroporation while the Figure 1 B shows the bar at around 10%. Supplementary Figure S1 also shows expression at 48h.

The y axis description of Figure 2 need to be improved.

Line 282: we tested them on three different

should be: we tested them in three different

Figure 3A – the axes titles of the FACS plots as well as population descriptions are barely readable

Comments on the Quality of English Language

The text is generally well written, some small  comments above.

Round 2

Reviewer 1 Report

Comments and Suggestions for Authors

Authors made many efforts to address referees' comments and substantially improved MS. However, their modifications raised several problems further and should not be ignored before going to publish. 

1# the usage of "construct" is misleading. Is it for a recombinant protein regime or an expression vector? It should be modified.

2# in "material and method", "12-21kmer of the S protein", referee doubts that it would be "12-21mer" for the length of peptides, right?

3# "a significant number of mutations", please specify how many is "significant"? and how comes to this threshold?

4# It is hard to understand the 5th tips about 19 epitopes within the corresponding MHC polymorphism coverage. "Randomly sampling all possible combinations", if someone talks about all combinations, how comes to "randomly sampling"? 

5# the introducing of DC-LAMP should be detailed narrated. There are plenty of previous reports on LAMP1, of which the chimeric strategy could marvelously enhance the antigen presentation and vaccine efficacies (10.1016/j.antiviral.2021.105141 or 10.3389/fimmu.2023.1253568). The choice of DC-LAMP and the recombinant strategy should be discussed. Especially, this would enroll self/auto-antigen.

Reviewer 3 Report

Comments and Suggestions for Authors

The manuscript has improved a lot. Good job.

Round 3

Reviewer 1 Report

Comments and Suggestions for Authors

Authors addressed most of the issues.